# New 12,23-Epoxydammarane Type Saponins Obtained from *Panax notoginseng* Leaves and Their Anti-Inflammatory Activity

**DOI:** 10.3390/molecules25173784

**Published:** 2020-08-20

**Authors:** Jingya Ruan, Ying Zhang, Wei Zhao, Fan Sun, Lifeng Han, Haiyang Yu, Lijie Wu, Yi Zhang, Tao Wang

**Affiliations:** 1Tianjin Key Laboratory of TCM Chemistry and Analysis, Tianjin University of Traditional Chinese Medicine, 10 Poyanghu Road, West Area, Tuanbo New Town, Jinghai District, Tianjin 301617, China; Ruanjy19930919@163.com (J.R.); zhaowei126123@126.com (W.Z.); sf18435165322@163.com (F.S.); hanlifeng_1@sohu.com (L.H.); wulj0816@163.com (L.W.); 2Institute of TCM, Tianjin University of Traditional Chinese Medicine, 10 Poyanghu Road, West Area, Tuanbo New Town, Jinghai District, Tianjin 301617, China; zyingtzy@163.com (Y.Z.); hyyu@tjutcm.edu.cn (H.Y.)

**Keywords:** *Panax notoginseng* leaves, 12,23-epoxydammarane-type saponin, notoginsenoside NL-J, RAW 246.7 cell, anti-inflammatory activity

## Abstract

Two new 12,23-epoxydammarane-type saponins, notoginsenosides NL-I (**1**) and NL-J (**2**), were isolated and identified from *Panax notoginseng* leaves through the combination of various chromatographies and extensive spectroscopic methods, as well as chemical reactions. Among them, notoginsenoside NL-J (**2**) had a new skeleton. Furthermore, the lipopolysaccharide (LPS)-induced RAW 264.7 macrophage model was used to identify the in vitro anti-inflammatory activity and mechanisms of compounds **1** and **2**. Both of them exerted strong inhibition on nitric oxide (NO) productions in a concentration-dependent manner at 1, 10, and 25 μM. Moreover, these two compounds significantly decreased the secretion of tumor necrosis factor-alpha (TNF-α), interleukin 6 (IL-6), cyclooxygenase-2 (COX-2), nuclear factor kappa-B (NF-κB/p65), and nitric-oxide synthase (iNOS) in LPS-activated RAW 264.7 cells.

## 1. Introduction

Inflammation is a protective immune response to external stimuli. However, if it is not controlled in time, excessive inflammation can promote the occurrence and development of inflammation-related diseases, such as type 2 diabetes, gout, atherosclerosis, enteritis, cancer, and other diseases [1]. The inflammatory response is closely related to the level of proinflammatory mediators, such as nitric-oxide (NO), inducible nitric-oxide synthase (iNOS), interleukin-6 (IL-6), tumor necrosis factor-alpha (TNF-α), cyclooxygenase-2 (COX-2), and nuclear factor kappa-B (NF-κB/p65), etc. According to the reported references, Abelson murine leukemia virus transformed macrophage RAW 264.7 cells will release TNF-α, IL-6, NF-κB, iNOS, COX-2, and other inflammatory factors after being stimulated by a lipopolysaccharide (LPS) [2,3,4].

*Panax notoginseng* (Burk.) F. H. Chen belongs to the *Panax* genus, Acanthopanax family. Its traditional pharmacological effect is to dissipate stasis and stop bleeding and disperse swelling and pain. It is mainly used for treating cardiovascular and cerebrovascular diseases, injuries, blood stasis, and swelling pain closely related to inflammation. To date, most of the current research subjects have focused on its root, owing to its extraordinary usage in inflammation-related diseases. Many researchers have verified its in vivo [5] and in vitro anti-inflammation abilities [6]. Owing to the gradual scarcity of its resources, the other parts of *P. notoginseng* have gradually become a research hotspot in recent years. In a previous study, we found that the dammarane triterpenoid saponins in *P. notoginseng* leaves had an excellent inhibitory effect on the production of nitric oxide from LPS-activated RAW 264.7 cells [7,8].

In this paper, through the combination of various chromatographies, extensive spectroscopic methods, and chemical reactions, the dammarane triterpenoid saponins with a 12,23-epoxydammarane skeleton in *P. notoginseng* leaves were studied. Then, the obtained compounds were screened for their anti-inflammatory activity and mechanisms by evaluating the level of proinflammatory mediators in LPS-induced RAW 264.7 macrophages.

## 2. Results

Various chromatographic methods, including D101 macroporous resin, silica gel, ODS, Sephadex LH-20 column chromatographies (CC), and preparative HPLC chromatorgraphy (pHPLC), spectrophotometric methods, and chemical reactions were used to gain saponins from the 50% ethanol extract of *P. notoginseng* leaves. Two new 20(*S*)-protopanaxadiol (PPD) type saponins, named notoginsenosides NL-I (**1**) and NL-J (**2**), were obtained from it (Figure 1).

Notoginsenoside NL-I (**1**) was obtained as a white powder with negative optical rotation ([α]_D_^25^ −10.4, MeOH). Its molecular formula, C_47_H_78_O_17_ (*m*/*z* 913.51648 [M − H]^−^; calcd. for C_47_H_77_O_17_, 913.51553) was measured on the negative-ion ESI-Q-Orbitrap MS. d-glucose and d-xylose were detected in its acid hydrolysis product by HPLC analysis [9]. The ^1^H and ^13^C NMR (Table 1) spectra showed signals for eight methyl groups at δ 0.82, 0.94, 1.01, 1.11, 1.31, 1.51, 1.68, and 1.83 (3H each, all s, H_3_-19, 18, 29, 30, 28, 21, 26, and 27), three oxygenated methines at δ 3.37 (1H, dd, *J* = 4.5, 11.5 Hz, H-3), 3.67 (1H, m, H-12), and 4.86 (1H, dd, *J* = 8.0, 10.0 Hz, H-23), one tri-substituted olefin group at δ 5.55 (1H, d, *J* = 8.0 Hz, H-24), two β-d-glucopyranosyl moieties at δ 4.97 (1H, d, *J* = 8.0 Hz, H-1′) and 5.12 (1H, d, *J* = 8.0 Hz, H-1″), and one β-d-xylopyranosyl group at δ 4.99 (1H, d, *J* = 7.5 Hz, H-1‴), which suggested that notoginsenoside NL-I (**1**) was a dammarane-type triterpene saponin. According to the correlations observed in its ^1^H ^1^H COSY spectrum, seven moieties, written in bold lines (Figure 2), were deduced. Meanwhile, in order to solve the problem of overlapping for the three glycosyl groups, a HSQC-TOCSY experiment was performed. In the HSQC-TOCSY spectrum, correlations were found between δ_H_ 4.97 (H-1′) and δ_C_ 71.9 (C-4′), 75.8 (C-2′), 78.9 (C-3′), 107.0 (C-1′); δ_H_ 4.43, 4.63 (H_2_-6′) and δ_C_ 63.1 (C-6′), 71.9 (C-4′), 78.4 (C-5′); δ_H_ 5.12 (H-1″) and δ_C_ 71.7 (C-4″), 75.3 (C-2″), 76.9 (C-5″), 78.8 (C-3″), 99.4 (C-1″); δ_H_ 4.40, 4.77 (H_2_-6″) and δ_C_ 70.9 (C-6″), 71.7 (C-4″), 76.9(C-5″); δ_H_ 4.99(H-1‴) and δ_C_ 71.2 (C-4‴), 78.3 (C-3‴), 74.9 (C-2‴), 106.5 (C-1‴); δ_H_ 3.71, 4.38 (H_2_-5‴) and δ_C_ 67.3 (C-5‴), 71.2 (C-4‴). In conjunction with the ^1^H ^1^H COSY and HSQC spectra, the spectroscopic data of the above-mentioned three glycosyls were assigned in detail. Moreover, the planar structure of **1** was determined by the long-range correlations displayed from H_3_-18 to C-7–9, C-14; H_3_-19 to C-1, C-5, C-9, C-10; H_3_-21 to C-17, C-20, C-22; H_3_-23 to C-12, C-24, C-25; H_3_-26 to C-24, C-25, C-27; H_3_-27 to C-24–26; H_3_-28 to C-3–5, C-29; H_3_-29 to C-3–5, C-28; H_3_-30 to C-8, C-13–15; H-1′ to C-3; H-1″ to C-20; H-1‴ to C-6″ in its HMBC experiment (Figure 2). The chemical shifts of protons and carbons in A and B rings resembled notoginsenoside NL-A_1_ [7], which suggested that the A and B rings of notoginsenoside NL-I (**1**) were identical to it. Moreover, the NOE correlations observed between δ_H_ 1.49 (H-9) and δ_H_ 3.67 (H-12); δ_H_ 3.67 (H-12) and δ_H_ 3.22 (H-17), 4.86 (H-23); δ_H_ 3.22 (H-17) and δ_H_ 4.86 (H-23) in the NOESY spectrum (Figure 2) indicated that H-12 and H-23 were α-orientation, like H-9, while the β-orientations of H-13 and 21-CH_3_ were confirmed by the cross peaks observed between δ_H_ 1.59 (H-13) and δ_H_ 0.94 (H_3_-18), 1.51 (H_3_-21). The aglycone of notoginsenoside NL-I (**1**) was the same as that of notoginsenoside LX [10] and was named (20*S*,23*R*)-3β,20-dihydroxy-12β,23-epoxy-dammar-24-ene.

Notoginsenoside NL-J (**2**) was obtained as a white powder with negative optical rotation ([α]_D_^25^ −10.0, MeOH). The ESI-Q-Orbitrap MS experiment result indicated that the molecular fomula of **13** was C_57_H_94_O_27_ (*m/z* 1209.59021 [M − H]^−^; calcd. for C_57_H_93_O_27_, 1209.58987). The IR spectrum showed that the absorption bands were assignable to hydroxyl (3356 cm^−1^), carbonyl (1699 cm^−1^), and ether (1071 cm^−1^) functions. Its ^1^H and ^13^C NMR (Table 1) spectra suggested the presence of three β-d-glucopyranosyl [δ 4.95 (1H, d, *J* = 8.4 Hz, H-1′), 5.06 (1H, d, *J* = 7.8 Hz, H-1‴′), and 5.54 (1H, d, *J* = 7.2 Hz, H-1″)] and two β-d-xylopyranosyl [δ 4.93 (1H, d, *J* = 7.8 Hz, H-1‴″), 5.41 (1H, d, *J* = 7.2 Hz, H-1‴)]. Fifty-seven signals showed in the ^13^C NMR spectrum. Except for the signals belonging to the above-mentioned glycosyl groups, twenty-nine remained for its aglycone. In the aglycone, one carbonyl appeared, but one methyl and one olefin disappeared compared with that of notoginsenoside NL-I (**1**). According to the proton and proton correlations observed in its ^1^H ^1^H COSY spectrum (Figure 2), nine moieties, written in bold lines, were denoted. Its planar structure was clarified according to the long-range correlations observed from H_3_-18 to C-7–9, C-14; H_3_-19 to C-1, C-5, C-9, C-10; H_3_-21 to C-17, C-20, C-22; H_3_-23 to C-12, C-24, C-25; H_3_-26 to C-24, C-25; H_3_-28 to C-3–5, C-29; H_3_-29 to C-3–5, C-28; H_3_-30 to C-8, C-13–15; H-1′ to C-3; H-1″ to C-2′; H-1‴ to C-2″; H-1‴′ to C-20; H-1‴″ to C-6‴′ (Figure 2), which suggested that it was a 27-demethyl PPD-type saponin. Finally, the NOE correlations displayed between H-9 and H-12; H-12 and H-17, H-23; H-17 and H-23; H-13 and H_3_-18, H_3_-21 in its NOESY spectrum denoted the configuration of the A–E ring of **2** was identical to that of **1**. Then, the structure of notoginsenoside NL-J (**2**) was elucidated, which was a new skeleton compound.

Moreover, the NO inhibitory effects of compounds **1** and **2** were examined in LPS-stimulated RAW 264.7 cells. Pretreatment of them at noncytotoxic concentration (25 μM) (Appendix A) decreased the NO production significantly compared with the control group, which indicated that they had the anti-inflammatory activity. Moreover, the inhibitory activities of compounds **1** and **2** were concentration-dependent at 1, 10, and 25 μM (Table 2).

As TNF-α, IL-6, COX-2, NF-κB, and iNOS were reported to be the major inflammatory cytokines, the Western blot method was used to study the anti-inflamatory mechanisms of compounds **1** and **2** by determining their expressions in LPS-induced RAW 264.7 cells. Compared with the normal group, LPS led to an obvious upregulation in the protein expressions of TNF-α, IL-6, COX-2, NF-κB, and iNOS. Both compounds **1** and **2** exhibited the inhibitory effects of these protein expressions in the cells (Figure 3 and Figure 4).

## 3. Discussion

According to the reference [11], the main triterpenoids in *P. notoginseng* had tetracyclic dammarane triterpene skeletons—the 12,23-epoxydammarane triterpenoids were not only rare in *P. notoginseng* but also in natural products. Until now, they were only found in *Panax* genus plants, such as *P. notoginseng* [12], *P. quinquefolium* [13,14], and *P. ginseng* [15], along with the *Gynostemma* genus plant, *G. pentaphyllum* [16].

Different from our previous research [7,8], a new 12,23-epoxydammarane skeleton has been found in this paper, and its superior in vitro anti-inflammatory activites, compared with the other compounds reported earlier [7,8], made it more promising to become the target of new anti-inflammatory drugs. What is more, the results of the Western blot assays provided more evidence for its further development.

## 4. Experimental

### 4.1. Experimental Procedures for Phytochemistry Study

#### 4.1.1. General Experimental Procedures

NMR spectra were determined on a Bruker Ascend 600 MHz and/or Bruker Ascend 500 MHz NMR spectrometer (Bruker BioSpin AG Industriestrasse 26 CH-8117, Fällanden, Switzerland), with tetramethylsilane as an internal standard. Negative-ion mode ESI-Q-Orbitrap MS were measured on a Thermo ESI-Q-Orbitrap MS mass spectrometer connected to the UltiMate 3000 UHPLC instrument via ESI interface (Thermo, Waltham, MA). Optical rotation, UV, and IR spectra were run on a Rudolph Autopol^®^ IV automatic polarimeter (l = 50 mm) (Rudolph Research Analytical, Hackettstown NJ, USA) and Varian 640-IR FT-IR spectrophotometer (Varian Australia Pty Ltd., Mulgrave, Australia), respectively.

CC was performed on macroporous resin D101 (Haiguang Chemical Co., Ltd., Tianjin, China), silica gel (48–75 μm, Qingdao Haiyang Chemical Co., Ltd., Qingdao, China), ODS (50 μm, YMC Co., Ltd., Tokyo, Japan), MCI gel CHP 20P (Mitsubishi Chemical Corporation, Osaka, Japan), and Sephadex LH-20 (Ge Healthcare Bio-Sciences, Uppsala, Sweden). A high performance liquid chromatography (HPLC) column, the Cosmosil 5C_18_-MS-II (4.6 mm i.d. × 250 mm, 5 µm, Nakalai Tesque, Inc., Tokyo, Japan), was used for analysis, and a Cosmosil 5C_18_-MS-II (20 mm i.d. × 250 mm, 5 µm, Nakalai Tesque, Inc., Tokyo, Japan) was used to separate the constituents.

#### 4.1.2. Plant Material

The leaves of *Panax notoginseng* (Burk.) F. H. Chen were collected from Shilin County, Kunming City, Yunnan Province, China, and identified by Dr. Wang Tao (Institute of Traditional Chinese Medicine, Tianjin University of Traditional Chinese Medicine). The voucher specimen was deposited at the Academy of Traditional Chinese Medicine of Tianjin University of traditional Chinese medicine (TCM).

#### 4.1.3. Extraction and Isolation

The dried leaves of *P. notoginseng* (8 kg) were extracted three times with 50% ethanol (EtOH) under reflux for 3 h, 2 h, and 2 h, successively. Evaporation of the solvent under reduced pressure provided the 50% EtOH extract (2.67 kg). Then, an aliquot (2.1 kg) of the 50% EtOH extract was subjected to D101 resin CC (H_2_O → 95% EtOH) to give H_2_O (760.0 g) and 95% EtOH eluted fraction (695.0 g), respectively.

The 95% EtOH eluate (150.0 g) was subjected to silica gel CC [CH_2_Cl_2_-MeOH (1:0 → 100:3 → 100:7 → 10:1 → 8:1 → 3:1 → 2:1 → 1:1 → 0:1, *v/v*)] to yield twelve fractions (Fr. 1–Fr. 12). Fraction 7 (30.0 g) was separated by MCI gel CHP 20P CC [MeOH-H_2_O (65:35 → 70:30 → 75:25 → 80:20 → 100:0, *v*/*v*)], and twelve fractions (Fr. 7-1–Fr. 7-12) were given. Fraction 7-6 (800.0 mg) was separated by pHPLC [MeOH-1% HAc (70:30, *v/v*), Cosmosil 5C_18_-MS-II column], and seven fractions (Fr. 7-6-1–Fr. 7-6-7) were obtained. Fraction 7-6-4 (179.5 mg) was purified by pHPLC [CH_3_CN-1% HAc (32:68, *v/v*), Cosmosil 5C_18_-MS-II column] to give notoginsenoside NL-I (**1**, 8.7 mg). Fraction 9 (15.0 g) was isolated by MCI gel CHP 20P CC [MeOH-H_2_O (60:40 → 70:30 → 80:20 → 100:0, *v/v*)], and ten fractions (Fr. 9-1–Fr. 9-10) were given. Among them, Fraction 9-3 (254.2 mg) was further separated by pHPLC [CH_3_CN-1% HAc (22:78, *v/v*), Cosmosil 5C_18_-MS-II column] to yield notoginsenoside NL-J (**2**, 3.2 mg).

*Notoginsenoside NL-I (***1***)*: White powder; [α]_D_^25^ −10.4 (*conc* 0.34, MeOH); IR *ν*_max_ (KBr) 3365, 2941, 2877, 1632, 1449, 1380, 1163, 1074, 1039 cm^−1^; ^1^H NMR (C_5_D_5_N, 500 MHz) spectroscopic data: δ 0.82, 1.50 (1H each, both m, overlapped, H_2_-1), [1.83 (1H, m, overlapped), 2.25 (1H, m), H_2_-2], 3.37 (1H, dd, *J* = 4.5, 11.5 Hz, H-3), 0.72 (1H, br. d, ca. *J* = 12 Hz, H-5), [1.35 (1H, m), 1.49 (1H, m, overlapped), H_2_-6], 1.19, 1.39 (1H each, both m, H_2_-7), 1.49 (1H, m, overlapped, H-9), 1.35, 1.93 (1H each, both m, H_2_-11), 3.67 (1H, m, H-12), 1.59 (1H, dd, *J* = 11.0, 11.0 Hz, H-13), [1.06 (1H, m), 1.48 (1H, m, overlapped), H_2_-15], 2.13, 2.32 (1H each, both m, H_2_-16), 3.22 (1H, dt, *J* = 4.0, 11.0 Hz, H-17), 0.94 (3H, s, H_3_-18), 0.82 (3H, s, H_3_-19), 1.51 (3H, s, H_3_-21), [2.26 (1H, dd, *J* = 10.0, 16.0 Hz), 2.85 (1H, d, *J* = 16.0 Hz), H_2_-22], 4.86 (1H, dd, *J* = 8.0, 10.0 Hz, H-23), 5.55 (1H, d, *J* = 8.0 Hz, H-24), 1.68 (3H, s, H_3_-26), 1.83 (3H, s, H_3_-27), 1.31 (3H, s, H_3_-28), 1.01 (3H, s, H_3_-29), 1.11 (3H, s, H_3_-30), 4.97 (1H, d, *J* = 8.0 Hz, H-1′), 4.06 (1H, dd, *J* = 7.5, 8.0 Hz, H-2′), 4.28 (1H, dd, *J* = 7.5, 9.0 Hz, H-3′), 4.25 (1H, m, overlapped, H-4′), 4.05 (1H, m, H-5′), [4.43 (1H, m), 4.63 (1H, br. d, ca. *J* = 12 Hz), H_2_-6′], 5.12 (1H, d, *J* = 8.0 Hz, H-1″), 3.94 (1H, dd, *J* = 8.0, 8.0 Hz, H-2″), 4.22 (1H, dd, *J* = 8.0, 9.0 Hz, H-3″), 4.17 (1H, dd, *J* = 9.0, 9.0 Hz, H-4″), 4.15 (1H, m, overlapped, H-5″), [4.40 (1H, dd, *J* = 6.0, 11.0 Hz), 4.77 (1H, br. d, ca. *J* = 11 Hz), H_2_-6″], 4.99 (1H, d, *J* = 7.5 Hz, H-1‴), 4.09 (1H, dd, *J* = 7.5, 8.5 Hz, H-2‴), 4.16 (1H, dd, *J* = 8.5, 9.0 Hz, H-3‴), 4.26 (1H, m, overlapped, H-4‴), [3.71 (1H, dd, *J* = 11.0, 11.0 Hz), 4.38 (1H, dd, *J* = 5.0, 11.0 Hz), H_2_-5‴]; ^13^C NMR (C_5_D_5_N, 125 MHz) spectroscopic data: see Table 1; ESI-Q-Orbitrap MS *m*/*z* 913.51648 [M − H]^−^ (calcd. for C_47_H_77_O_17_, 913.51553).

*Notoginsenoside NL-J (***2***)*: White powder; [α]_D_^25^ −10.0 (*conc* 0.16, MeOH); IR *ν*_max_ (KBr) 3356, 2934, 2877, 1699, 1568, 1413, 1167, 1071, 1041 cm^−1^; ^1^H NMR (C_5_D_5_N, 600 MHz) spectroscopic data: δ 0.63, 1.29 (1H each, both m, H_2_-1), 1.80, 2.15 (1H each, both m, H_2_-2), 3.29 (1H, dd, *J* = 2.4, 12.0 Hz, H-3), 0.66 (1H, br. d, ca. *J* = 10 Hz, H-5), 1.37, 1.46 (1H each, both m, overlapped, H_2_-6), [1.19 (1H, m), 1.37 (1H, m, overlapped), H_2_-7], 1.37 (1H, m, overlapped, H-9), 1.22, 1.77 (1H each, both m, H_2_-11), 3.57 (1H, m, H-12), 1.49 (1H, dd, *J* = 11.4, 11.4 Hz, H-13), 1.02, 1.43 (1H each, both m, H_2_-15), 2.09 (2H, m, H_2_-16), 3.02 (1H, dt, *J* = 3.6, 11.4 Hz, H-17), 0.89 (3H, s, H_3_-18), 0.76 (3H, s, H_3_-19), 1.46 (3H, s, H_3_-21), [1.98 (1H, dd, *J* = 9.0, 15.0 Hz), 2.77 (1H, br. d, ca. *J* = 15 Hz), H_2_-22], 4.62 (1H, m, overlapped, H-23), 2.86 (2H, m, H_2_-24), 2.21 (3H, s, H_3_-26), 1.26 (3H, s, H_3_-27), 1.10 (3H, s, H_3_-28), 0.97 (3H, s, H_3_-29), 4.95 (1H, d, *J* = 8.4 Hz, H-1′), 4.15 (1H, m, overlapped, H-2′), 4.38 (1H, m, overlapped, H-3′), 4.11 (1H, m, overlapped, H-4′), 3.99 (1H, m, H-5′), [4.38 (1H, m, overlapped), 4.61 (1H, br. d, ca. *J* = 11 Hz), H_2_-6′], 5.54 (1H, d, *J* = 7.2 Hz, H-1″), 4.21 (1H, m, overlapped, H-2″), 4.30 (1H, dd, *J* = 9.0, 9.0 Hz, H-3″), 4.20 (1H, m, overlapped, H-4″), 3.89 (1H, m, H-5″), [4.37 (1H, m, overlapped), 4.51 (1H, br. d, ca. *J* = 11 Hz), H_2_-6″], 5.41 (1H, d, *J* = 7.2 Hz, H-1‴), 4.12 (1H, m, overlapped, H-2‴), 4.15 (1H, m, overlapped, H-3‴), 4.16 (1H, m, overlapped, H-4‴), [3.72 (1H, dd, *J* = 10.2, 10.2 Hz), 4.34 (1H, m, overlapped), H_2_-5‴], 5.06 (1H, d, *J* = 7.8 Hz, H-1″″), 3.96 (1H, dd, *J* = 7.8, 8.0 Hz, H-2″″), 4.26 (1H, m, overlapped, H-3″″), 4.20 (1H, m, overlapped, H-4″″), 4.12 (1H, m, H-5″″), [4.32 (1H, m, overlapped), 4.77 (1H, br. d, ca. *J* = 10 Hz), H_2_-6″″], 4.93 (1H, d, *J* = 7.8 Hz, H-1″″), 4.06 (1H, dd, *J* = 7.8, 9.0 Hz, H-2″″), 4.17 (1H, m, overlapped, H-3″″), 4.26 (1H, m, overlapped, H-4″″), 3.70, 4.37 (1H each, both m, overlapped, H_2_-5″″); ^13^C NMR (C_5_D_5_N, 150 MHz) spectroscopic data: see Table 1; ESI-Q-Orbitrap MS *m*/*z* 1209.59021 [M − H]^−^ (calcd. for C_57_H_93_O_27_, 1209.58987).

*Acid Hydrolysis of***1***and***2**: Compounds **1** and **2** were hydrolyzed by 1M HCl and aqueous layers were obtained by using the reported method [9]. As the results shown, d-glucose and d-xylose were detected in their acid hydrolysate. All of the sugars were confirmed by comparison of *t*_R_ with the relative authentic samples [*t*_R_: 9.4 min (d-xylose), and 12.4 min (d-glucose); all of them showed positive optical rotations].

### 4.2. Experimental Procedures for Bioassay

The MTT, nitrite levels, and the Western blot assay measurements of **1** and **2**, as well as the statistical analysis of them, were conducted using the method reported previously [3,10].

## 5. Conclusions

In the process of investigating anti-inflammatory dammarane-type triterpene saponins, two new 12,23-epoxydammarane-type saponins, notoginsenosides NL-I (**1**) and NL-J (**2**), were obtained from the 50% ethanol extract of *P. notoginseng* leaves. Notoginsenoside NL-J (**2**) was determined to have a new skeleton.

Furthermore, both of the two compounds were found to inhibit LPS-induced NO release in a concentration-dependent manner at 1, 10, and 25 μM. It is worth mentioning that notoginsenoside NL-J (2) still possessed strong biological activity at 1 μM. In order to make the mechanism of their anti-inflammatory activities clear, Western blot assays were conducted. As a result, both of them were found to inhibit the LPS-induced protein expression of TNF-α, IL-6, COX-2, NF-κB, and iNOS in the NF-κB signaling pathway. According to what has been reported previously [17], as these two compounds exhibited NF-κB downregulated activities, their anti-inflammation mechanisms may be related to inhibiting the expression of inflammatory cytokines. If it could directly suppress the activation of the NF-κB pathway, more Western blot assays should be conducted.

## Figures and Tables

**Figure 1 molecules-25-03784-f001:**
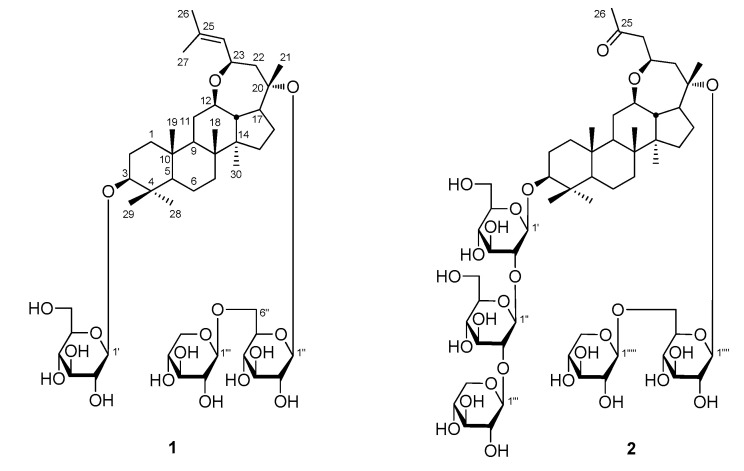
The structures of compounds (**1**) and (**2**).

**Figure 2 molecules-25-03784-f002:**
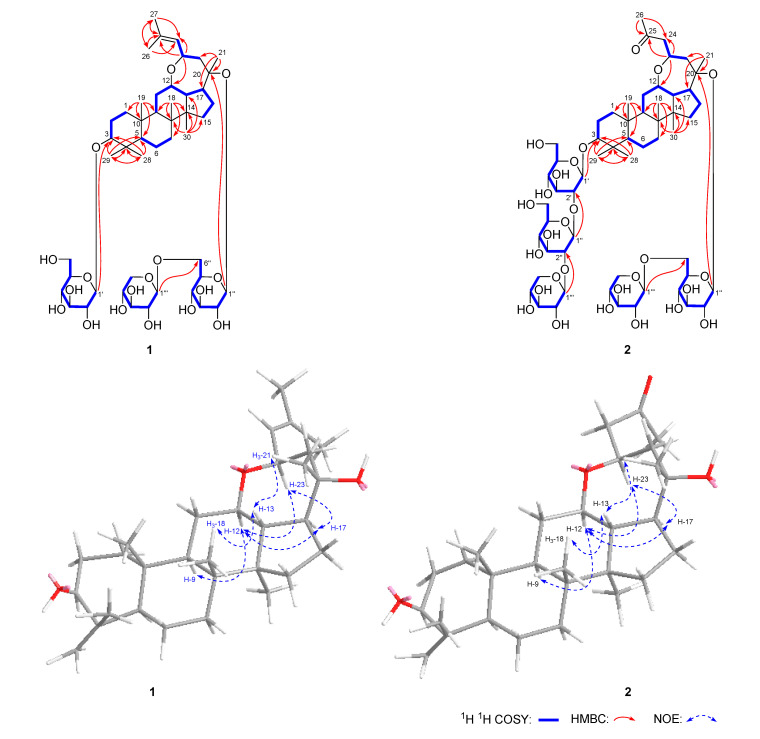
Main ^1^H ^1^H COSY, HMBC, and NOE correlations of (**1**) and (**2**)**.**

**Figure 3 molecules-25-03784-f003:**
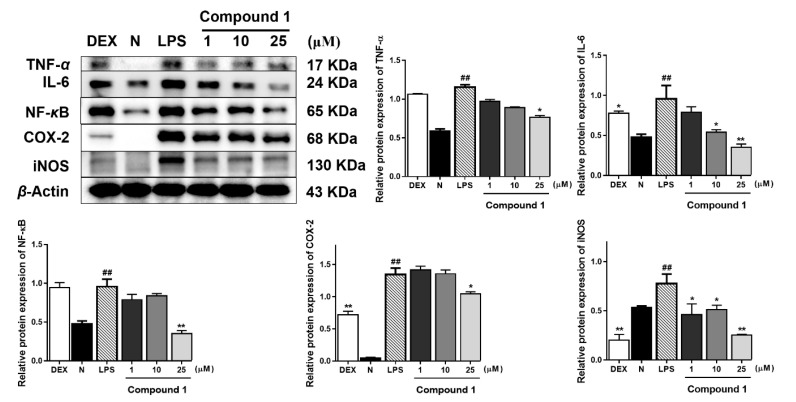
Inhibitory effects of compound **1** on the protein expression of tumor necrosis factor-alpha (TNF-α), interleukin-6 (IL-6), cyclooxygenase-2 (COX-2), nuclear factor kappa-B (NF-κB), and inducible nitric-oxide synthase (iNOS) in RAW 264.7 cells. Normal: normal group without LPS, DEX, and other tested samples. Values represent the mean ± SEM of three determinations. * *p* < 0.05; ** *p* < 0.01; (Differences between compound-treated group and control group). ^##^
*p* < 0.01 (Differences between LPS-treated group and control group). *n* = 3.

**Figure 4 molecules-25-03784-f004:**
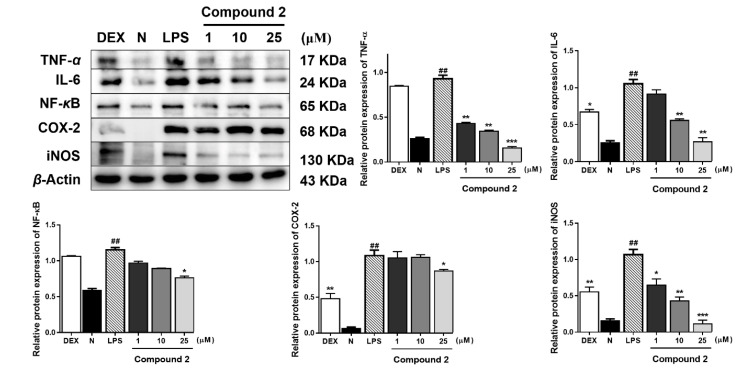
Inhibitory effects of compound **2** on the protein expression of TNF-α, IL-6, COX-2, NF-κB, and iNOS in RAW 264.7 cells. Normal: normal group without LPS, DEX, and other tested samples. Values represent the mean ± SEM of three determinations. * *p* < 0.05; ** *p* < 0.01; *** *p* < 0.001; (Differences between compound-treated group and control group). ^##^
*p* < 0.01 (Differences between LPS-treated group and control group). *n* = 3.

**Table 1 molecules-25-03784-t001:** ^13^C NMR data of **1** and **2** in C_5_D_5_N.

No.	1	2	No.	1	2	No.	1	2
1	39.4	39.2	21	24.7	24.1	5″	76.9	77.8
2	26.8	26.8	22	52.1	50.2	6″	70.9	62.9
3	88.7	88.8	23	72.6	71.0	1‴	106.5	106.5
4	39.7	39.7	24	129.2	51.6	2‴	74.9	76.0
5	56.3	56.4	25	131.2	207.2	3‴	78.3	77.8
6	18.4	18.4	26	25.7	30.5	4‴	71.2	70.8
7	35.2	35.1	27	18.9		5‴	67.3	67.4
8	39.8	39.7	28	28.1	28.0	1″″		99.0
9	50.6	50.5	29	16.8	16.7	2″″		75.4
10	37.1	37.0	30	17.0	16.8	3″″		78.7
11	30.1	29.9	1′	107.0	104.8	4″″		71.6
12	79.6	80.0	2′	75.8	83.0	5″″		76.7
13	49.8	49.4	3′	78.9	78.7	6″″		70.5
14	51.3	51.2	4′	71.9	71.2	1‴″		106.1
15	32.6	32.5	5′	78.4	78.4	2‴″		74.9
16	25.6	25.2	6′	63.1	63.0	3‴″		78.4
17	46.5	47.0	1″	99.4	103.2	4‴″		71.2
18	15.5	15.4	2″	75.3	84.6	5‴″		67.2
19	16.5	16.5	3″	78.8	78.0			
20	82.0	81.6	4″	71.7	71.8			

**Table 2 molecules-25-03784-t002:** Inhibitory effects of **1** and **2** on nitric oxide (NO) production in RAW 264.7 cells.

**No**	**NRC (%)**	**NRC (%)**		
		25 μM	10 μM	1 μM
Normal	3.2 ± 0.6			
Control	100 ± 4.2			
DEX	75.3 ± 2.6 ***			
1		48.0 ± 9.1 ***	90.6 ± 3.2 ***	104.4 ± 8.0
2		42.4 ± 8.6 ***	80.3 ± 6.3 ***	81.9 ± 10.7 ***

Positive control: Dexamethasone (Dex). Nitrite relative concentration (NRC): percentage of control group (set as 100%). Values represent the mean ± SD of four determinations. *** *p* < 0.001 (Differences between compound-treated group and control group). n = 6. Final concentration was 25, 10, 1 μM for compounds **1** and **2** and 1.0 μg/mL for positive control (Dex), respectively. The concentration of nitrite produced in the LPS control group was 50 μM.

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
