# Peer review of "New 12,23-Epoxydammarane Type Saponins Obtained from Panax notoginseng Leaves and Their Anti-Inflammatory Activity"

_molecules, 2020, doi:10.3390/molecules25173784_

Round 1

Reviewer 1 Report

Potentially interesting paper needs significant revision. The main problems are as follows:

1) Differences between previous findings on same/similar subject need to be described and discussed.

2) Using Results & Discussion together is not a good idea, in this particular case there actually is no discussion and only description of results. I would recommend to split it into two section and completely re-write the Discussion.

3) Conclusion – what the data really mean? Does the author expect any follow up experiments?

Reviewer 2 Report

Article: Molecules-891477

Title: New 12,23-Epoxydammarane Type Saponins Obtained from Panax notoginseng Leaves and Their Anti-Inflammatory Activity

The manuscript from Ruan et al. reports on the isolation and identification of two new 12,23-epoxydammarane type saponins from Panax notoginseng Leaves. Moreover, the anti-inflammatory properties of these compounds have been investigated in LPS-stimulated RAW264.7, a widely used mouse model of inflammation. The experiments of both chemical and biological sections have been well conducted and the results are fair. However, some drawbacks are present and listed below.  

Major:

1) The introduction is rather short and does not give enough information on the state of the art of this research topic. The authors should include some more references and discuss them properly. Please consider:

Wang et al., ACS Chemical Neuroscience, 2017, 8, 548-557; Plastina et al., Nutrients, 2018, 10, 783; Plastina et al., Food Chemistry, 2019, 279, 105-113, as other references regarding the in vitro evaluation of anti-inflammatory properties in a mouse model;

Chan et al., International Journal of Biological Macromolecules, 2020, 155, 376-385; Zhou et al., Biomedicine & Pharmacotherapy, 2020, 127, 110139; Guo et al., Journal of Agricultural and Food Chemistry, 2020, 68, 6835-6844; Ma et al., Journal of Agricultural and Food Chemistry, 2020, 68, 5327-5338; Shang et al., Journal of Ginseng Research, 2020, 44, 405-412; Chao et al., Journal of Pharmaceutical and Biomedical Analysis, 2020, 182, 113127; Jang et al., Journal of Asian Natural Product Research (DOI: 10.1080/10286020.2020.1731801); Wang et al., Food Chemistry, 2020, 303, 125394;  as other references regarding (bioactive) compounds present in Panax notoginseng, including other saponins with anti-inflammatory properties.

2) Figure 3 is not fully visible in the manuscript. Moreover, as authors express NO inhibition as percentage with respect to LPS control, they should give an indication of the concentration of nitrite produced in the LPS control in the figure caption.

Minor:

Please consider that the correct publication year of reference #4 is 2020.

Abstract, line 24 (and possibly throughout the manuscript): “µm” should be changed into “µM”

Page 4, line 119 (and possibly throughout the manuscript): “dose” should be changed into “concentration”, as this is not an in vivo study.

Round 2

Reviewer 1 Report

The changes improved the quality of the manuscript.

Reviewer 2 Report

The authors addressed properly all the concerns raised from the previous version and changed the manuscript accordingly. I suggest the acceptance of the manuscript in its revised form as is.